# Perception of Healthcare Providers during the COVID-19 Pandemic: A Mixed Method Survey in an Integrated Healthcare Delivery System in Saudi Arabia

**DOI:** 10.3390/ijerph192416676

**Published:** 2022-12-12

**Authors:** Ali Faris Alamri, Fahad Khamees Alomari, Amir Moustafa Danash, Maram Talal Aljoudi, Asmahan Issa Almasharqa, Ahmed Metwally Almasloot, Reem M. Alwhaibi, Mohamed Mossad Hasan, Uzma Zaidi

**Affiliations:** 1King Abdullah Bin Abdulaziz University Hospital, Princess Nourah bint Abdulrahman University, Riyadh 13415, Saudi Arabia; 2Prince Sultan Military Hospital, Taif 1087, Saudi Arabia; 3Lincoln County Hospital, Lincolnshire NG318UJ, UK; 4Rehabilitation Sciences Department, College of Health and Rehabilitation Sciences, Princess Nourah bint Abdulrahman University, Riyadh 13415, Saudi Arabia; 5Feinberg School of Medicine, Northwestern University, Chicago, IL 60645, USA; 6Department of Health Sciences, College of Health and Rehabilitation Sciences, Princess Nourah bint Abdulrahman University, Riyadh 13415, Saudi Arabia

**Keywords:** telemedicine, tele-clinics, perception of use, perception of ease of use, behavioral intention, adoption, survey, COVID-19, Saudi Arabia

## Abstract

During the COVID-19 pandemic, telemedicine was broadly adopted for patient care. Considering this experience, it is crucial to understand the perceptions of teleclinic healthcare professionals. In Saudi Arabia, telemedicine literature was restricted to physicians working in government and private hospitals. This study examined perceptions in relation to telemedicine among physicians and other healthcare professionals practicing in Saudi Arabian military hospitals in the Taif region. During COVID-19, telemedicine was implemented in military hospitals; consequently, this study assists in evaluating introduced practices and the perceptions of health professionals regarding these new practices. A quantitative, descriptive, correlational, and cross-sectional study was undertaken on healthcare professionals (*N* = 153). Twenty (20) items based on standardized measures were used to collect data using an online questionnaire. The measures contained three subscales: perceived usefulness, perceived ease, and behavioral intention. It was hypothesized that the perception of teleclinic usefulness and ease score by healthcare providers would be significantly correlated with behavioral intention. Descriptive statistics for mean, frequency, and standard deviation, as well as a Pearson correlation coefficient and regression analysis, were conducted to assess the relationship and predictive association between variables. In addition, a focus group discussion was organized to collect information directly from healthcare professionals. Most of the participants were approximately 40 years of age, Saudi Nationals (63%), medical specialists (62%), and were involved in teleclinic practices before the COVID-19 pandemic (60%). The reliability of all three scales was determined to be acceptable (α = 0.75–0.91). Perceived usefulness and perceived ease were shown to be significantly correlated with behavioral intention (r = 0.877, *p* = 0.05). In addition, the regression analysis indicated that perceived usefulness and perceived ease are predictors of the behavioral intention (R^2^ = 0.777, F (2,152) = 261.76, *p* = 0.001) of teleclinic practices among healthcare professionals. The positive perception of telemedicine integration in healthcare systems revealed by this study is a major catalyst for continuous adoption. On the other hand, certified telemedicine platforms, on-the-job training, Internet of things, and a flexible approach are required to find opportunities and enhancements in telemedicine interactions.

## 1. Introduction

Telemedicine has been widely embraced as a means of caring for COVID-19 patients as well as patients suffering from other acute and chronic conditions [1,2,3,4]. Telemedicine is defined by the WHO as “the provision of healthcare services at a distance with communication conducted between healthcare providers seeking clinical guidance and support from other healthcare providers (provider-to-provider telemedicine); or conducted between remote healthcare users seeking health services and healthcare providers (client-to-provider telemedicine)” [5]. In the US, during 2020 it was reported that there was a ten-fold increase in the use of telemedicine [6]. As viral spread can occur from asymptomatic patients as well as patients in recovery, telemedicine became an essential social distancing tool for the safety of healthcare workers and the community [1,4,6,7,8]. Historically, the adoption of telemedicine was slow, but the COVID-19 pandemic galvanized the alignment of infrastructure, technology, and governmental will to deliver safe and high-quality care via telemedicine [1,4]. While the COVID-19 pandemic spurred the adoption of telemedicine, the benefits associated with its use make it highly unlikely that telemedicine will dissipate after the pandemic fades [1,2,3]. Telemedicine is less costly and more convenient, and it encourages patients to seek care early and thus avoid disease progress due to delayed care [7]. In addition, telemedicine improves access to high-quality care for patients in geographically remote areas [7,8].

Guided by Saudi Vision 2030, leaders in Saudi Arabia have invested in telemedicine to enhance healthcare access, particularly in rural/remote areas [9,10,11,12]. The Saudi Ministry of Health (MOH) launched the Saudi Telemedicine Network (STN) to cover all healthcare facilities [13]. At the onset of the pandemic, the Saudi health sector resorted to mobile applications and teleclinics to ensure access while maintaining public health measures [14,15,16]. Before the COVID-19 pandemic, studies in Saudi Arabia showed a slow adoption of telemedicine, partly due to the lack of acceptance among healthcare workers [11,17,18,19,20]. As healthcare workers’ exposure to telemedicine has significantly increased throughout the pandemic, it is essential to understand their perceptions based on this experience [21]. While other studies examined the perception of telemedicine in Saudi Arabia during the COVID-19 pandemic, they were confined only to physicians and did not include those working in military hospitals [21,22]. This study aims to investigate the perception and behavior of telemedicine among physicians and other healthcare workers practicing in military hospitals in the Taif region of Saudi Arabia. Therefore, it was hypothesized that there is a relationship between the score of tele-clinicians on perceived usefulness (PI), perceived ease of use (PEU), and behavioral intention (BI).

## 2. Materials and Methods

### 2.1. Study Design and Participants

A cross-sectional, descriptive, correlational, and quantitative study was conducted at the Armed Forces Hospital, Taif region from December 2020 to June 2021. These hospitals constitute an integrated healthcare delivery system in the western region of Saudi Arabia that consists of a tertiary hospital, an acute care hospital, a community hospital, a rehabilitation center, and a psychiatric center. The Armed Forces Hospital adopted teleclinics during the COVID-19 pandemic to allow access while maintaining control of the pandemic. The target population for the study comprised all physicians, nurses, and allied health professionals that use teleclinics and work at the Armed Forces Hospital, Taif region. Therefore, all male and female physicians and health professionals, regardless of their age, nationality, and work experience were included. Email invitations were sent to the target group followed by an email reminder. The sample size was calculated by the OpenEpi calculator using the formula:n = [DEFF × Np(1 − p)]/[(d^2^/Z^2^_1-α/2_ × (N − 1) + p × (1 − p)]
where N was 700, the margin of error was 5%, and the sample size was 249 [23]. A total of 318 participants responded to the email. However, only 153 survey forms were correctly filled.

The characteristics of participants are presented in Table 1. The gender-wise participation was almost equal. However, most of the specialists were from the medical field (62%). The majority (88%) of healthcare workers had been serving for more than two years. Most of the record-keeping (83%) was already available through the electronic system. Most of the healthcare workers (60%) were involved in teleclinic practices before the COVID-19 pandemic.

### 2.2. Measures

The first section of the questionnaire was related to demographic data, work-related variables, experience working with teleclinics before the COVID-19 pandemic, and the type of medical records used at the hospital. The second part was developed by adapting various validated questionnaires [24,25,26,27,28,29,30]. This questionnaire measured three constructs: perceived usefulness (PU), perceived ease of use (PEU), and behavioral intention (BI). Perceived usefulness consisted of ten (10) items that were measured by items such as “using a teleclinic increases my job productivity.” Perceived ease of use contained four (4) items that were measured by items such as “a teleclinic makes me feel comfortable most of the time.” Behavioral intention was measured by six (6) items such as “maintaining a teleclinic post-pandemic is highly important”. Responses ranged from strongly disagree to strongly agree (1–5) on the Likert scale. A consent form was provided to respondents to ensure confidentiality, privacy, and volunteer participation. Moreover, ethical approval was obtained from the Institutional Review Board (CREC 2020-025) of the Prince Sultan Military Hospital, Taif Region.

### 2.3. Statistical Analysis

Descriptive statistics were conducted in the form of mean and standard deviation for numeric variables. The alpha coefficient was calculated for the three scales of perception of usefulness (PU), perception of ease of usefulness (PEU), and behavioral intention (BI). Furthermore, the Pearson correlation coefficient was conducted to measure the relationship between PU and PEU on BI after conducting the Kolmogorov–Smirnov test for the normal distribution of scores. A regression analysis was conducted to measure the predictive association of PU and PEU on BI. SPSS V. 26 for Windows was used for the analysis, and *p*-value < 0.05 was considered significant.

### 2.4. Focus Group Procedure

Moreover, four focus group discussions (*N* = 48) were conducted by experts to understand the perception of teleclinic healthcare professionals regarding the strengths, barriers, and challenges of their practices. The survey participants who provided consent to take part in the focus group discussion were contacted, and formal arrangement measures were taken. Each focus group discussion was conducted for 90–100 min. Each focus group consisted of twelve (12) healthcare professionals. A written consent form was provided to all focus group members explaining the confidentiality and privacy of data. Moreover, the focus group facilitator verbally explained the ground rules of participation and the purpose of conducting the group discussions. The six (6) questions were presented to focus group participants. Transcriptions were then prepared for analysis purposes. Qualitative data were categorized into main themes and sub-themes. Themes are available on request (https://osf.io/mk9tj/).

## 3. Results

For the measurement of perceptions of teleclinic health professionals, three scales were developed based on standardized scale items [23,24,25,26,27,28,29]. However, the reliability and inter consistency of items were measured by Cronbach’s alpha coefficient. The overall alpha coefficient showed that the PU scale had a good level of reliability (α = 0.91); PEU presented an acceptable range of reliability (α = 0.752), whereas the BI scale was found to be at a good level (α = 0.835). Table 2 presents Cronbach’s alpha coefficient for the PU, PEU, and BI scales.

Table 3 presents the Kolmogorov–Smirnov test to measure the normal distribution of the score on the PU, PEU, and BI scores by medical, surgical, and allied health professionals. The scores are all higher than the significance level of 0.5, which allowed for the further application of inferential statistics.

The Pearson correlation coefficient was conducted to measure the relationship between perceptions of behavioral intention. Table 4 shows a strong positive correlation of PU and PE with BI (r = 0.877, r = 0.571, *p* < 0.5). Both perceptions of PU and PEU were also found to be positively correlated (r = 0.724, *p* < 0.5) with each other.

A multiple regression was carried out to investigate whether PU and PEU could significantly predict BI among healthcare professionals for teleclinics (Table 5). The results of the regression indicated that the model explained 77.7% of the variance and that the model was a significant predictor of BI, F (2,152) = 261.76, *p* = 0.001. PU significantly contributed to the model (B = 0.551, *p* < 0.01), whereas PEU also significantly contributed (B = −0.175, *p* < 0.01), although inversely.

The final predictive model is:BI score = 5.321 + (0.551 × PU) + (−0.175 × PEU)

## 4. Qualitative Analysis for Focus Group Discussion

The qualitative data were divided into six (6) main themes. Table 6 presents the description and sub-theme distribution of the qualitative data.

## 5. Discussion

The results address the central question of the study; perceived usefulness (PU) and perceived ease of use (PEU) are not only positively related to behavioral intention (BI) but also predict it. Moreover, among tele-clinicians, regardless of their specialty, PU and PE can result in the variability of BI occurrence. These study findings are essential as more than half the participants experienced teleclinics for the first time during the COVID-19 pandemic. Moreover, the integrated healthcare delivery system was introduced within the Armed Forces Hospital setup for the first time. Therefore, we had to involve the maximum number of healthcare practitioners serving at the Armed Forces Hospital of the Taif region to gain in-depth knowledge about their telehealth practice. Overall, five various clinical setups were contacted; namely, a community hospital and tertiary, acute, rehabilitation, and psychiatric centers. Furthermore, the focus group discussion also revealed perceptions, challenges, and valuable suggestions by tele-clinicians to improve the interaction and sustainability of telemedicine practice.

The COVID-19 pandemic swiftly accelerated the otherwise reluctant adoption of telemedicine, which doubtlessly will persist as a standard modality in healthcare delivery [1,2,3,4,25,31]. While a multitude of factors play a role in the sustainability of telemedicine, the positive perception of integrating telemedicine in healthcare systems, as shown in our study and other local and international studies, is a strong facilitator for sustained adoption [16,21,22,23,24,30,31]. On the other hand, it was difficult to integrate telemedicine with some specialized fields, where direct communication was unavoidable or practitioners were not trained [2,3,7,31,32,33]. However, there were perceptions such as that PU and PEU supported cultivating strong BI among practitioners. The results of this study incorporate previous studies [21,22,23,24,33]. PU and PEU were simultaneously found to have a predictive association with BI. The analysis further revealed that if a health practitioner perceives teleclinics as useful, it will increase BI. In contrast, if the practitioner is not comfortable with the use of teleclinics, it will decrease BI.

In the literature, the existing scales measured various aspects of knowledge, perception, and attitudes, but the main concern of this study was based upon perceptions of health providers toward teleclinics during the pandemic. Therefore, two main variables of perceptions, including PU and PEU, were selected on the basis of the TAM-3 model [27]. Perceived usefulness perception is concerned with the sense of productivity, effectiveness, saving time, relevance to specialty, significance, and support provided by information and communications technology (ICT) and organizations [21,27,29]. On the other hand, PEU is more related to being comfortable with use, requiring less effort, and enjoyable usage of technology [27,30]. Finally, the outcome measure of BI depends upon a willingness to continue, integrate, and expand the practice [21,27,28,29]. All the scales were found to be reliable and acceptable to a good level. Moreover, it confirmed that the adopted measures were culturally sound.

The focus group discussion provided a rich description within six areas of strength, relevance, challenge, perceived ease of use, procedural enhancement, and sustainability. To get the group to open up, participants were asked what aspects of a teleclinic they consider to be key elements. Most of the participants agreed that they found the teleclinic practice to be time-saving, stress-free, and helping them to deliver health services without any delay [14,21,25]. Some participants identified that the teleclinic was also a money-saver for patients. The second aspect was the relevance of teleclinics with the specialty. The concept of relevance directly relates to PU [27]. The prevalent feeling among physicians was satisfaction with their specialty and teleclinic. However, most of the surgeons and allied health professionals described difficulties. Most of the participants emphasized the difficulty of using ICT and video calls to resolve these issues.

At this point, it is necessary to understand the third theme of challenges faced by health providers using teleclinics [12]. Numerous categories of response appeared in response to this theme. Some participants highlighted the difficulty of the absence of physical examinations, diagnostic data of the patient, or difficulty in the management of cases. Meanwhile, others placed more emphasis on challenges related to communication, patients’ awareness of the teleclinic ensuring the privacy of patients, and difficult populations, including children and the elderly. Some health providers experienced difficulty with ICT-related issues and the wrong data entry of patient phone numbers [19]. On the fourth theme of PEU, we asked how teleclinics can become more comfortable. Three major categories were identified: peer support, organizational support, and technological support. Most respondents suggested the use of better ICT and IoT [12,16,17]. Additionally, respondents shared that only patients could call in some hospital settings, so they recommended initiating a two-way calling system.

On the fifth theme of procedural enhancement strategies, suggestions were shared for training at the individual level, the use of IoT in a technological context, and a quality management system for the betterment of organizational context [27]. Participants were asked to suggest one regulation/practice or system that supports the sustainability of teleclinics in the health care system. The prevalent need was related to video calls, call recording, time scheduling, and cooperation of professionals. The application of international and national systems was suggested for sustainable teleclinics.

In conclusion, medical, surgical, and allied healthcare providers presented a positive perception of telemedicine toward the behavioral intention to practice. Our results have important clinical and research implications.

### Limitations and Implications

This study has some limitations. The questionnaire was introduced in English rather than the native language of the user, which might have induced a bias based on participants’ interpretation of the questionnaire. However, clear instructions were provided in Arabic. Moreover, a focus group discussion was conducted to overcome this gap. For future studies, the Arabic-translated version of our validated scales can be used. Finally, this study involved healthcare providers working in military hospitals in one region, which limits the generalizability of the results to other military and non-military hospitals. This study has provided reliable scales that can be used for future research. Most of the research was focused on the knowledge, attitude, and behavior of tele-clinicians but our study provided the link between perception and intended behavior. Moreover, suggestions gathered by the focus group discussion can be utilized to improve individual, organizational, and technological contexts for sustainable practices.

## 6. Conclusions

This study was conducted to measure the perceptions and behavioral intention of telemedicine among healthcare providers who worked in teleclinics during the COVID-19 pandemic in an integrated delivery system in Saudi Arabia. The results showed a positive relationship between perceived usefulness and perceived ease of use with the behavioral intention of teleclinic providers. Moreover, we found high agreement among participants as to the value of telemedicine in saving time and money and the importance of integrating telemedicine within the hospital system. The results of the focus group discussion highlighted the need for the latest technology, i.e., IoT and ICT, as well as international systems to improve the quality of teleclinic interaction.

## Figures and Tables

**Table 1 ijerph-19-16676-t001:** Characteristics of participants.

Characteristics of Participants		M (SD)
Age in years		39.93 (9.27)
		Frequency (Percentage)
Gender	Male	73(48)
	Female	80 (52)
Nationality	Saudi	97 (63)
	Non-Saudi	56 (37)
Specialty	Medical	95 (62)
	Surgical	20 (13)
	Health Alliance	38 (25)
Position	Consultant	44 (29)
	Specialist	85 (56)
	Resident	24 (15)
Years in hospital or department	Less than two years	19 (12)
	More than two years	134 (88)
A medical record in the hospital or department	Paper	3 (2)
	Electronic	83 (54)
	Mixed	67 (44)
Have you ever been involved in providing telecare for patients through teleclinics before the COVID-19 pandemic	No	92 (60)
	Yes	61 (40)

**Table 2 ijerph-19-16676-t002:** Reliability of perceived usefulness, perceived ease of use, and behavioral intention for the teleclinic scale.

S.	Subscales	α
1.	Perceived Usefulness	0.91
2.	Perceived Ease of Use	0.752
3.	Behavioral Intention	0.835

**Table 3 ijerph-19-16676-t003:** Test of normality.

Variables	Specialty		Kolmogorov–Smirnov ^a^	
		Statistic	df	Sig.
Perceived Usefulness		0.089	95	0.061
	Medical	0.097	20	0.200 *
	Surgical	0.101	38	0.200 *
Perceived Ease of Usefulness	Health Alliance	0.091	95	0.056
	Medical	0.169	20	0.138
	Surgical	0.134	38	0.084
Behavioral Intention	Health Alliance	0.127	95	0.061
	Medical	0.145	20	0.200 *
	Surgical	0.116	38	0.200 *

* This is a lower bound of true significance. ^a^. Lilliefors significance correction.

**Table 4 ijerph-19-16676-t004:** Correlation matrix of PU, PEU, and BI.

Variables	Perceived Usefulness	Perceived Ease of Usefulness	Behavioral Intention
Perceived Usefulness	1		
Perceived Ease of Usefulness	0.724 **	1	
Behavioral Intention	0.877 **	0.571 **	1

** *p* > 0.01.

**Table 5 ijerph-19-16676-t005:** Predictors of BI.

Variables	B	SE B	Beta
(Constant)	5.321 **	0.86	
Perceived Usefulness	0.551 **	0.03	0.974
Perceived Ease of Usefulness	−0.175 **	0.07	−0.135
R^2^	0.777 **		
F	261.768 **		

** *p* < 0.01.

**Table 6 ijerph-19-16676-t006:** Main and sub-themes of the qualitative data.

Main Theme	Description	Sub-Theme	Example
1. Strengths	It includes all the perceived beneficial aspects of teleclinics that healthcare professionals consider to be key elements.	Time saving	“Saves time due to easy access.”
		Money saving	“It is cost-effective.”
		Stress and Risk-free	“It lowers the risk of infection for me and the patient.”
		Continuous service being provided	“To serve remote patients.”
2. Relevance	The perception of the relevance of teleclinics with the specialty of medical, surgical, and allied health professionals.	Workable	“To update patient about results to adjust treatment.”
		Not workable	“Not all patients.”
		Possible solutions	“Need screen, video call to see the patient.”
3. Challenges	It includes challenges related to the treatment issue, specific population, and organization-related issues that become an obstacle in the process of teleclinics.	Physical examination	“Cannot physically assess the patient when it is required.”
		Diagnostic data	“Need X-ray for final diagnosis.”
		Difficulty in Management of cases	“Effective With only follow-up easy cases.”
		Communication	“Communication gap, difficulty to picture out exercises in mind.”
		Privacy	“Making sure of patient identity and reliability.”
		Population	“Difficult in youngsters.”
		Patients’ awareness of teleclinic	“Demanding patients, problems for other specialties.”
		IT related issues	“Bad ICT service.”
		Data entry	“Lack of correct contact number.”
4. Ease of Use Perception	It emphasizes factors that can enhance ease of teleclinic use by peer, organizational, technical, and communication support.	Peer Support	“Cooperation”.
		Organizational Support	“More malleability in timing.”
		Technical Support	“Add video call facility.”
		Communication patterns	“To call the patient rather than the patient calling us.”
5. Procedural enhancement strategies	It includes suggestions given to improve teleclinic interactions between the doctor and patient, technical, and organizational resources	Individual Context: Training	“Training is needed.”
		Technological Context: Internet of Things (IoT)	“Interactive Media needed.”
		Organizational Context	“Better organization of clinics.”
		Individual Context: Training	“Training is needed.”
6. Sustainability Requirement	It includes suggestions related to regulations or systems that could support the integration of teleclinics in the health care system.	Regulation/Practice	“Should be recorded with high confidentiality.”
		System	“Improving information technology.”

## Data Availability

Not applicable.

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
