# Peer review of "Perception of Healthcare Providers during the COVID-19 Pandemic: A Mixed Method Survey in an Integrated Healthcare Delivery System in Saudi Arabia"

_ijerph, 2022, doi:10.3390/ijerph192416676_

Round 1
Reviewer 1 Report
This study needs extensive improvement in its analysis part.
Other comments are as follows:
Title
Perception of Tele clinicians During Covid-19 Pandemic: A Survey Study of Healthcare Providers in An Integrated Healthcare Delivery System in Saudi Arabia
Better change it as,
Perception of healthcare providers during Covid-19 pandemic: A mixed method survey in an integrated healthcare delivery system in Saudi Arabia
Abstract:
Line 20; Covid-19 is not an epidemic....
Lines 30-1; Not an appropriate sentence for a completed research rather can be used in a synopsis, "It was expected....."
Lines 31-33: The sentence is not complete, better review
Line 41, IoT has been used for 1st time in manuscript so use full form here
Introduction
Reference 5 is not relevant as it is about Framework for the Implementation of a Telemedicine Service and does not show definition of telemedicine. You can find definition of telemedicine on various other WHO documents, e.g. https://apps.who.int/iris/rest/bitstreams/1346306/retrieve
Line 54 should mention time frame for this forecast in USA
Methodology
Line 85; mention healthcare delivery system
Line 86; what is meant by acute hospital?
There is written throughout the manuscript that Focus Group was organized, better write it as Focus Group Discussion throughout
Line 150; 700 participants were emailed, please correct it
Results
No need of putting so much tables just on reliability, it appears that this manuscript was about reliability and not on the perceptions of healthcare providers on telemedicine
Instead the manuscript results should focus on likert scale analysis
Author Response
Dear Reviewer,
The authors are grateful for the comments and suggestions of the reviewer. The manuscript has gone through an extensive English language correction by a professional editing service. We highly appreciate the encouragement and valuable suggestions from reviewers. The detailed response is in attachment.
Kind Regards.

Reviewer 2 Report
This paper aims to examine the attitudes of healthcare providers in the integrated healthcare delivery system in Saudi Arabia toward telemedicine. The analysis is detailed and recommended for acceptance after revision. But the following issues remain.
1、In the Abstract section, the number of participants, the response rate, and a description of the participants' characteristics should give.
2、The sample in section 2.1 should be described by gender, age, and other characteristics.
3、In section 2.3, “Alpha Coefficient was calculated for the three scales respectively Per-ception of Usefulness (PU) Perception of Ease of Usefulness (PEU) and Behavioural In-dention (BI).” How Alpha Coefficient is calculated, please give relevant literature or explanation.
4、In the results section, “The overall alpha coefficient showed that the PU scale showed a good level of relia- 136 bility (α=0.91).” α = 0.91 was not found in Table 1.
5、“A total of 700 healthcare providers were emailed an invitation to participate in the 150 survey, and the response rate was more than 123% (318).” please check if the response rate is miscalculated.
6、Table 4 should be placed in Section 2.1.
7、Please consider placing sections 4.2, 4.3, 4.4, and 4.5 in the appendix.
Author Response
Dear Reviewer,
The authors are grateful for the comments and suggestions of both reviewers. The manuscript has gone through an extensive English language correction by a professional editing service. We are grateful for appreciating the significance and scope of the article and suggesting us to improve the quality of the article. The detailed response is in attachment.
Kind Regards.

Round 2
Reviewer 1 Report
The manuscript is in much better shape now.
Just see the title of table 4, no need to write regression analysis in the title of this table. Also this table is copied as such from SPSS output, please revisit.
The manuscript otherwise is good to be accepted for publication
Author Response
Dear Reviewers,
The authors are grateful for the comments and suggestions of both reviewers. We have corrected the suggested issues in the article.
Kind Regards

Reviewer 2 Report
-
I recommend acceptance of this manuscript.
Author Response

(The authors gave the same response as above.)
